# Zero-shot Knowledge Transfer via Adversarial Belief Matching

**Paul Micaelli**
University of Edinburgh
{paul.micaelli}@ed.ac.uk

**Amos Storkey**
University of Edinburgh
{a.storkey}@ed.ac.uk

## Abstract

Performing knowledge transfer from a large teacher network to a smaller student is a popular task in modern deep learning applications. However, due to growing dataset sizes and stricter privacy regulations, it is increasingly common not to have access to the data that was used to train the teacher. We propose a novel method which trains a student to match the predictions of its teacher without using any data or metadata. We achieve this by training an adversarial generator to search for images on which the student poorly matches the teacher, and then using them to train the student. Our resulting student closely approximates its teacher for simple datasets like SVHN, and on CIFAR10 we improve on the state-of-the-art for few-shot distillation (with 100 images per class), despite using no data. Finally, we also propose a metric to quantify the degree of belief matching between teacher and student in the vicinity of decision boundaries, and observe a significantly higher match between our zero-shot student and the teacher, than between a student distilled with real data and the teacher. Code is available at:
https://github.com/polo5/ZeroShotKnowledgeTransfer

## 1 Introduction

Large neural networks are ubiquitous in modern deep learning applications, including computer vision (He et al., 2015), speech recognition (van den Oord et al., 2016) and natural language understanding (Devlin et al., 2018). While their size allows learning from big datasets, it is a limitation for users without the appropriate hardware, or for internet-of-things applications. As such, the deep learning community has seen a focus on model compression techniques, including knowledge distillation (Ba and Caruana, 2014; Hinton et al., 2015), network pruning (Li et al., 2016; Han et al., 2016) and quantization (Gupta et al., 2015; Hubara et al., 2016).

These methods typically rely on labeled data drawn from the training distribution of the model that needs compressed. Distillation does so by construction, and pruning or quantization need to fine-tune networks on training data to get good performance. We argue that this is a strong limitation because pretrained models are often released without training data, an increasingly common trend that has been grounds for controversy in the deep learning community (Radford et al., 2019). We identify four main reasons why datasets aren't released: privacy, property, size, and transience. Respective examples include Facebook's DeepFace network trained on four million confidential user images (Taigman et al., 2014), Google's Neural Machine Translation System trained on internal datasets (Wu et al., 2016) and regarded as intellectual property, the JFT-300 dataset which contains 300 million images across more than 18k classes (Sun et al., 2017), and finally the field of policy distillation in reinforcement learning (Rusu et al., 2016), where one requires observations from the original training environment which may not exist anymore. One could argue that missing datasets can be emulated with proxy data for distillation, but in practice that is problematic for two reasons. First, there is a correlation between data that is not publicly released and data that is hard to emulate, such as medical

datasets of various diseases Burton et al. (2015), or datasets containing several thousand classes like JFT. Secondly, it has been shown in the semi supervised setting that out-of-distribution samples can cause significant performance drop when used for training (Oliver et al., 2018).

As such, we believe that a focus on zero-shot knowledge transfer is justified, and our paper makes the following contributions: 1) we propose a novel adversarial algorithm that distills a large teacher into a smaller student without any data or metadata, 2) we show its effectiveness on two common datasets, and 3) we define a measure of belief match between two networks in the vicinity of one's decision boundaries, and demonstrate that our zero-shot student closely matches its teacher.

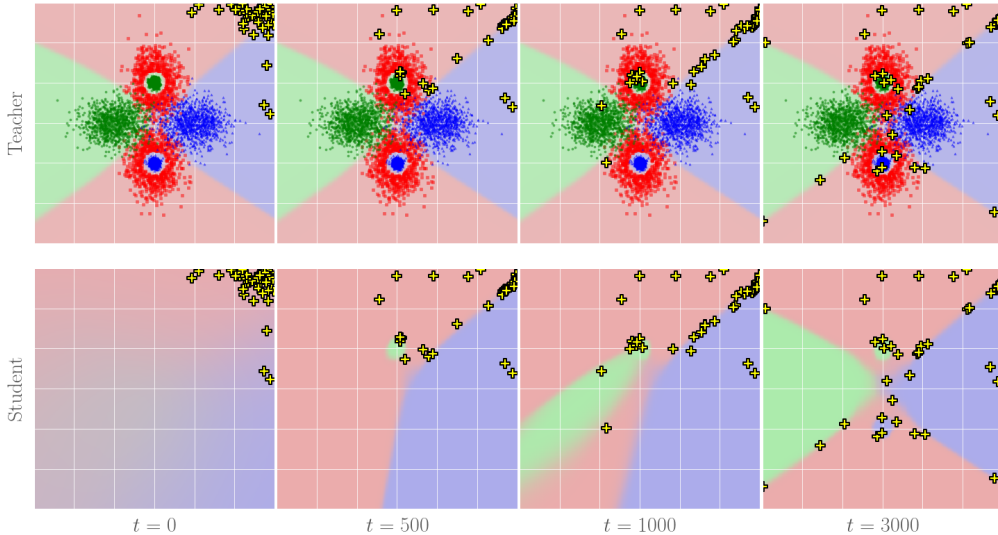

Figure 1: A simplified version of our method on a three-class toy problem. The teacher and student decision boundaries are shown in the top and bottom rows respectively. The knowledge transfer unfolds left to right and never uses the real data points, which are shown for visualization purposes. We initialize random pseudo points (yellow/black crosses) away from the data manifold, and train them to maximize the KL divergence between the student and teacher. At the same time we train the student to achieve the opposite. Note how pseudo points use the decision boundaries as channels to explore the input space, but can also explore regions away from them such as the two isolated green and blue pockets. After a few steps the student and teacher decision boundaries are indistinguishable.

## 2 Related work

**Inducing point methods and dataset distillation.** Inducing point methods (Snelson and Ghahramani, 2005) were introduced to make Gaussian Processes (GP) more tractable. The idea is to choose a set of inducing points that is smaller than the input dataset, in order to reduce inference cost. While early techniques used a subset of the training data as inducing points (Candela and Rasmussen, 2005), creating pseudo data using variational techniques was later shown more efficient (Titsias, 2009). Dataset distillation is related to this idea, and uses a bi-level optimization scheme to learn a small subset of pseudo images, such that training on those images yields representations that generalize well to real data (Wang et al., 2018). The main difference with these methods and ours is that we do not need the training data to generate pseudo data, but we rely on a pretrained teacher instead.

**Knowledge distillation.** The idea of using the outputs of a network to train another was first proposed by Buciluǎ et al. (2006) as a way to compress a large ensemble into a single network, and was later made popular by Ba and Caruana (2014) and then Hinton et al. (2015), who proposed smoothing the teacher's probability outputs. Since then, the focus has mostly been on improving distillation efficiency by designing better students (Romero et al., 2015; Crowley et al., 2018), or getting small performance gains with extra loss terms, such as attention transfer (AT) (Zagoruyko and Komodakis, 2016a). Since the term *knowledge distillation* (KD) has become intertwined with the loss function introduced by Hinton et al. (2015), we refer to our task more generally as zero-shot knowledge transfer (KT) for clarity.

**Privacy attacks.**   There are a few approaches to making privacy attacks that are related to our method. In model extraction (Tramèr et al., 2016) we have access to the probability predictions of a black-box model and the aim is to extract an equivalent model. The limitation of black-box access makes this task harder and often limited to simple datasets. In model inversion (Fredrikson et al., 2014, 2015) we have white-box access to a pretrained model, and we wish to recreate training images from the weights alone. This is a different aim from that of our task, because we wish to produce images that are relevant for training regardless of whether or not they resemble the training data.

**Zero-shot learning.**   In zero-shot learning (Larochelle et al., 2008; Socher et al., 2013), we are typically given training images with labels and some additional intermediate semantic representation $T$, such as textual descriptions. The task is then to classify images at test time that are represented in $T$ but whose classes were never observed during training. In our model, the additional intermediate information can be considered to be the teacher, but none of the classes are formally observed during training because no samples from the training set are used.

**Zero and few-shot distillation.**   More recently, the relationship between data quantity and distillation performance has started being addressed. In the few-shot setting, Li et al. (2018) obtain a student by pruning a teacher, and align both networks with 1x1 convolutions using a few samples. Kimura et al. (2018) distill a GP to a neural network by adversarially learning pseudo data. In their setting however, the teacher itself has access to little data and is added to guide the student. Concurrent to our work, Ahn et al. (2019) formulated knowledge transfer as variational information distillation (VID) and to the best of our knowledge obtained state-of-the-art in few-shot performance. We show that our method gets better performance than they do even when they use an extra 100 images per class on CIFAR-10. The zero-shot setting has been a lot more challenging: most methods in the literature use some type of metadata and are limited to simple datasets like MNIST. Lopes et al. (2017) showed that training images could be reconstructed from a record of the teacher's training activations, but in practice releasing training activations instead of data is unlikely. Concurrent to our work, Nayak et al. (2019) synthesize pseudo data from the weights of the teacher alone and use it to train a student in zero-shot. Their model is not trained end-to-end, and on CIFAR-10 we obtain a performance 17% higher than they do for a comparable sized teacher and student.

## 3   Zero-shot knowledge transfer

### 3.1   Algorithm

Let $T(\boldsymbol{x})$ be a pretrained teacher network, which maps some input image $\boldsymbol{x}$ to a probability vector $\boldsymbol{t}$. Similarly, $S(\boldsymbol{x};\theta)$ is a student network parameterized by weights $\theta$, which outputs probability vector $\boldsymbol{s}$. Let $G(\boldsymbol{z};\phi)$ be a generator parameterized by weights $\phi$, which produces pseudo data $\boldsymbol{x}_p$ from a noise vector $\boldsymbol{z} \sim \mathcal{N}(\boldsymbol{0}, \boldsymbol{I})$. The main loss function we use is the forward Kullback–Leibler (KL) divergence between the outputs of the teacher and student networks on pseudo data, namely $D_{KL}(T(\boldsymbol{x}_p) \parallel S(\boldsymbol{x}_p)) = \sum_i \boldsymbol{t}_p^{(i)} \log(\boldsymbol{t}_p^{(i)}/\boldsymbol{s}_p^{(i)})$ where $i$ corresponds to image classes.

Our zero-shot training algorithm is described in Algorithm 1. For $N$ iterations we sample one batch of $\boldsymbol{z}$, and take $n_G$ gradient updates on the generator with learning rate $\eta$, such that it produces pseudo samples $\boldsymbol{x}_p$ that maximize $D_{KL}(T(\boldsymbol{x}_p) \parallel S(\boldsymbol{x}_p))$. We then take $n_S$ gradient steps on the student with $\boldsymbol{x}_p$ fixed, such that it matches the teacher's predictions on $\boldsymbol{x}_p$. The idea of taking several steps on the two adversaries has proven effective in balancing their relative strengths. In practice we use $n_S > n_G$, which gives more time to the student to match the teacher on $\boldsymbol{x}_p$, and encourages the generator to explore other regions of the input space at the next iteration.

### 3.2   Extra loss functions

Using the forward KL divergence as the main loss function encourages the student to spread its density over the input space and gives non-zero class probabilities for all images. This high student entropy is a vital component to our method since it makes it hard for the generator to fool the student too easily; we observe significant drops in student test accuracy when using the reverse KL divergence or Jensen–Shannon divergence. In practice, many student-teacher pairs have similar block structures,

and so we can add an attention term to the student loss:

$$\mathcal{L}_S = D_{KL}(T(\boldsymbol{x}_p) \,||\, S(\boldsymbol{x}_p)) + \beta \sum_l^{N_L} \left\| \frac{\boldsymbol{f}(A_l^{(t)})}{\left\| \boldsymbol{f}(A_l^{(t)}) \right\|_2} - \frac{\boldsymbol{f}(A_l^{(s)})}{\left\| \boldsymbol{f}(A_l^{(s)}) \right\|_2} \right\|_2 \tag{1}$$

where $\beta$ is a hyperparameter. We take the sum over some subset of $N_L$ layers. Here, $A_l^{(t)}$ and $A_l^{(s)}$ are the teacher and student activation blocks for layer $l$, both made up of $N_{A_l}$ channels. If we denote by $\boldsymbol{a}_{lc}$ the $c$th channel of activation block $A_l$, then we use the spatial attention map $\boldsymbol{f}(A_l) = (1/N_{A_l}) \sum_c \boldsymbol{a}_{lc}^2$ as suggested by the authors of AT (Zagoruyko and Komodakis, 2016a). We don't use attention for the generator loss $\mathcal{L}_G$ because it makes it too easy to fool the student.

Many other loss terms were investigated but did not help performance, including sample diversity, sample consistency, and teacher or student entropy (see Supplementary Material 1). These losses seek to promote properties of the generator that already occur in the plain model described above. This is an important difference with competing models such as that of Kimura et al. (2018) where authors must include hand designed losses like carbon copy memory replay (which freezes some pseudo samples in time), or fidelity (Dehghani et al., 2017).

---

**Algorithm 1:** Zero-shot KT (Section 3.1)

**pretrain:** $T(\cdot)$
**initialize:** $G(\cdot; \phi)$
**initialize:** $S(\cdot; \theta)$

**for** $1, 2, ..., N$ **do**
$\quad \boldsymbol{z} \sim \mathcal{N}(\boldsymbol{0}, \boldsymbol{I})$
$\quad$ **for** $1, 2, ..., n_G$ **do**
$\quad\quad \boldsymbol{x}_p \leftarrow G(\boldsymbol{z}; \phi)$
$\quad\quad \mathcal{L}_G \leftarrow -D_{KL}(T(\boldsymbol{x}_p) \,||\, S(\boldsymbol{x}_p))$
$\quad\quad \phi \leftarrow \phi - \eta \dfrac{\partial L_G}{\partial \phi}$
$\quad$ **end**

$\quad$ **for** $1, 2, ..., n_S$ **do**
$\quad\quad \mathcal{L}_S \leftarrow D_{KL}(T(\boldsymbol{x}_p) \,||\, S(\boldsymbol{x}_p))$
$\quad\quad \theta \leftarrow \theta - \eta \dfrac{\partial L_S}{\partial \theta}$
$\quad$ **end**
$\quad$ decay $\eta$
**end**

---

**Algorithm 2:** Compute transition curves of networks A and B, when stepping across decision boundaries of network A (Section 4.4)

**pretrain:** $net_A$
**pretrain:** $net_B$

**for** $\boldsymbol{x} \in \boldsymbol{X}_{test}$ **do**
$\quad i_A \equiv$ class of $\boldsymbol{x}$ according to $net_A$
$\quad i_B \equiv$ class of $\boldsymbol{x}$ according to $net_B$
$\quad$ **if** $i_A = i_B = i$ **then**
$\quad\quad \boldsymbol{x}_0 \leftarrow \boldsymbol{x}$
$\quad\quad$ **for** $j \neq i$ **do**
$\quad\quad\quad \boldsymbol{x}_{adv} \leftarrow \boldsymbol{x}_0$
$\quad\quad\quad$ **for** $1, 2, ..., K$ **do**
$\quad\quad\quad\quad \boldsymbol{y}_A, \boldsymbol{y}_B \leftarrow net_A(\boldsymbol{x}_{adv}), net_B(\boldsymbol{x}_{adv})$
$\quad\quad\quad\quad \boldsymbol{x}_{adv} \leftarrow \boldsymbol{x}_{adv} - \xi \dfrac{\partial \mathcal{L}_{CE}(\boldsymbol{y}_A, j)}{\partial \boldsymbol{x}_{adv}}$
$\quad\quad\quad\quad$ **save:** $\boldsymbol{y}_A, \boldsymbol{y}_B$
$\quad\quad\quad$ **end**
$\quad\quad$ **end**
$\quad$ **end**
**end**

---

### 3.3 Toy experiment

The dynamics of our algorithm is illustrated in Figure 1, where we use two layer MLPs for both the teacher and student, and learn the pseudo points directly (no generator). These are initialized away from the real data manifold, because we assume that no information about the dataset is known in practice. During training, pseudo points can be seen to explore the input space, typically running along decision boundaries where the student is most likely to match the teacher poorly. At the same time, the student is trained to match the teacher on the pseudo points, and so they must keep changing locations. When the decision boundaries between student and teacher are well aligned, some pseudo points will naturally depart from them and search for new high teacher mismatch regions, which allows disconnected decision boundaries to be explored as well.

### 3.4 Potential conceptual concerns

Here we address a number of potential conceptual concerns when dealing with the application of this approach to higher dimensional input spaces.

**Focus.** The first potential concern is that $G(\boldsymbol{z}; \phi)$ is not constrained to produce real or bounded images, and so it may prefer to explore the larger region of space where the teacher has not been trained. Assuming that the teacher's outputs outside of real images is irrelevant for the classification task, the student would never receive useful signal. In practice we observe that this assumption does not hold. On MNIST for instance, preliminary experiments showed that a simple student can achieve 90% test accuracy when trained to match a teacher on random noise. On more diverse datasets like CIFAR-10, we observe that uniform noise is mostly classified as the same class across networks (see Supplementary Material 2). This suggests that the density of decision boundaries is smaller outside of the real image manifold, and so $G(\boldsymbol{z}; \phi)$ may struggle to fool the student in that space due to the teacher being too predictable.

**Adversaries.** Another potential concern is that $G(\boldsymbol{z}; \phi)$ could simply iterate over adversarial examples, which in this context corresponds to images that are all very similar in pixel space and yet are classified differently by the teacher. Here we refer the reader to recent work by Ilyas et al. (2019), who isolate adversarial features and show that they are enough to learn classifiers that generalize well to real data. The bottom line is that non-robust features, in a human sense, still contain most of the task-relevant knowledge. Therefore, this suggests that our generator can produce adversarial examples (in practice, we observe that it does) while still feeding useful signal to the student.

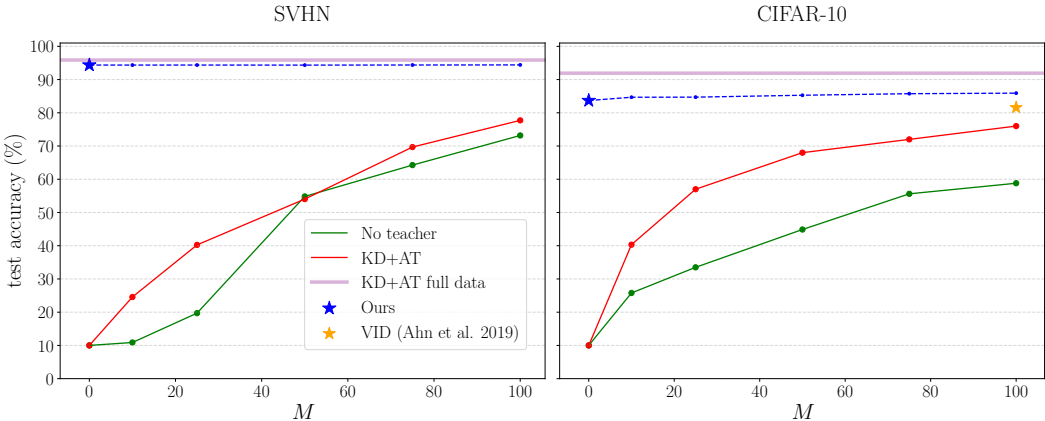

Figure 2: Performance of our model for a WRN-40-2 teacher and WRN-16-1 student, on the SVHN and CIFAR-10 datasets, for $M$ images per class. We compare this to the student when trained from scratch (no teacher), the student when trained with knowledge distillation and attention transfer from the teacher (KD+AT), and the performance of Ahn et al. (2019) (81.59% for $M = 100$). Our method reaches $83.69 \pm 0.58\%$ without using any real data, and increases to $85.91 \pm 0.24\%$ when finetuned with $M = 100$ images per class.

## 4 Experiments

In the few-shot setting, researchers have typically relied on validation data to tune hyperparameters. Since our method is zero-shot, assuming that validation data exists is problematic. In order to demonstrate zero-shot KT we thus find coarse hyperparameters in one setting (CIFAR-10, WRN-40-2 teacher, WRN-16-1 student) and use the same parameters for all other experiments of this paper. In practice, we find that this makes the other experiments only slightly sub optimal, because our method is very robust to hyperparameters and dataset change: in fact, we find that halving or doubling most of our hyperparameters has no effect on student performance. For each experiment we run three seeds and report the mean with one standard deviation. When applicable, seeds cover networks initialization, order of training images, and the subset of $M$ images per class in few-shot.

## 4.1 CIFAR-10 and SVHN

We focus our experiments on two common datasets, SVHN (Netzer et al., 2011) and CIFAR-10 (Krizhevsky, 2009). SVHN contains over 73k training images of 10 digits taken from house numbers in Google Street images. It is interesting for our task because most images contain several digits, the ground truth being the most central one, and so ambiguous images are easily generated. CIFAR-10 contains 50k training images across 10 classes, and is substantially more diverse than SVHN, which makes the predictions of the teacher harder to match. For both datasets we use WideResNet (WRN) architectures (Zagoruyko and Komodakis, 2016b) since they are ubiquitous in the distillation literature and easily allow changing the depth and parameter count.

Our distillation results are shown in Figure 2 for a WRN-40-2 teacher and WRN-16-1 student, when using $\mathcal{L}_S$ as defined in Equation 1. We include the few-shot performance of our method as a comparison, by naively finetuning our zero-shot model with $M$ samples per class. As baselines we show the student performance when trained from scratch (no teacher supervision), and the student performance when trained with both knowledge distillation and attention transfer, since that was observed to be better than either techniques alone. We also plot the equivalent result of VID (Ahn et al., 2019); to the best of our knowledge, they are the state-of-the-art in the few-shot setting at the time of writing. On CIFAR-10, we obtain a test accuracy of $83.69 \pm 0.58\%$ if we use the student loss described in Equation 1, which is $2\%$ better than VID's performance when it uses an extra $M = 100$ images per class. By finetuning our model with $M = 100$ our accuracy increases to $85.91 \pm 0.24\%$, pushing the previous few-shot state-of-the-art by more than $4\%$. If we do not use attention ($\beta = 0$) we observe an average drop of $2\%$ across all the architectures in Table 1.

Using all the same settings on SVHN yields a test accuracy of $94.06 \pm 0.27\%$. This is quite close to $95.88 \pm 0.15\%$, the accuracy obtained when using the full 73k images during KD+AT distillation, even though the hyperparameters and generator architecture were tuned on CIFAR-10. This shows that our model can be used on new datasets without needing a hyperparameter search every time, which is desirable for zero-shot tasks where validation data may not be available. If hyperparameters are tuned on SVHN specifically, our zero-shot performance is on par with full data distillation.

**Implementation details.** For the zero-shot experiments, we choose the number of iterations $N$ to match the one used when training the teachers on SVHN and CIFAR-10 from scratch, namely 50k and 80k respectively. For each iteration we set $n_G = 1$ and $n_S = 10$. We use a generic generator with only three convolutional layers, and our input noise $\mathbf{z}$ has 100 dimensions. We use Adam (Kingma and Ba, 2015) with cosine annealing, with an initial learning rate of $2 \times 10^{-3}$. We set $\beta = 250$ unless otherwise stated. For our baselines, we choose the same settings used to train the teacher and student in the literature, namely SGD with momentum 0.9, and weight decay $5 \times 10^{-4}$. We scale the number of epochs such that the number of iterations is the same for all $M$. The initial learning rate is set to 0.1 and is divided by 5 at $30\%, 60\%$, and $80\%$ of the run.

Table 1: Zero shot performance on various WRN teacher and student pairs for CIFAR-10. Our zero-shot technique ($M = 0$) is on par with KD+AT distillation when it's performed with $M = 200$ images per class. Teacher scratch and student scratch are trained with $M = 5000$. We report mean and standard deviation over 3 seeds.

| Teacher (# params) | Student (# params) | Teacher scratch | Student scratch | KD+AT $M = 200$ | Ours $M = 0$ |
|---|---|---|---|---|---|
| WRN-16-2 (0.7M) | WRN-16-1 (0.2M) | $93.97 _{+0.11}$ | $91.04 _{+0.04}$ | $\mathbf{84.54} _{\mathbf{+0.21}}$ | $82.44 _{+0.21}$ |
| WRN-40-1 (0.6M) | WRN-16-1 (0.2M) | $93.18 _{+0.08}$ | $91.04 _{+0.04}$ | $\mathbf{81.71} _{\mathbf{+0.25}}$ | $79.93 _{+1.11}$ |
| WRN-40-2 (2.2M) | WRN-16-1 (0.2M) | $94.73 _{+0.02}$ | $91.04 _{+0.04}$ | $81.25 _{+0.67}$ | $\mathbf{83.69} _{\mathbf{+0.58}}$ |
| WRN-40-1 (0.6M) | WRN-16-2 (0.7M) | $93.18 _{+0.08}$ | $93.97 _{+0.11}$ | $85.74 _{+0.47}$ | $\mathbf{86.60} _{\mathbf{+0.56}}$ |
| WRN-40-2 (2.2M) | WRN-16-2 (0.7M) | $94.73 _{+0.02}$ | $93.97 _{+0.11}$ | $86.39 _{+0.33}$ | $\mathbf{89.71} _{\mathbf{+0.10}}$ |
| WRN-40-2 (2.2M) | WRN-40-1 (0.6M) | $94.73 _{+0.02}$ | $93.18 _{+0.08}$ | $\mathbf{87.35} _{\mathbf{+0.12}}$ | $86.60 _{+1.79}$ |

## 4.2 Architecture dependence

While our model is robust to the choice of hyperparameters and generator, we observe that some teacher-student pairs tend to work better than others, as is the case for few-shot distillation. We

compare our zero-shot performance with $M = 200$ KD+AT distillation across a range of network depths and widths. The results are shown in Table 1. The specific factors that make for a good match between teacher and student are to be explored in future work. In zero-shot, deep students with more parameters don't necessarily help: the WRN-40-2 teacher distills $3.1\%$ better to WRN-16-2 than to WRN-40-1, even though WRN-16-2 has less than half the number of layers, and a similar parameter count than WRN-40-1. Furthermore, the pairs that are strongest for few-shot knowledge distillation are not the same as for zero-shot. Finally, note that concurrent work by Nayak et al. (2019) obtains $69.56\%$ for $M = 0$ on CIFAR-10, despite using a hand-designed teacher/student pair that has more parameters than the WRN-40-1/WRN-16-2 pair we use. Our method thus yields a $17\%$ improvement, but this is in part attributed to the difference of efficiency in the architectures chosen.

## 4.3 Nature of the pseudo data

Samples from $G(\boldsymbol{z}; \phi)$ during training are shown in Figure 3. We notice that early in training the samples look like coarse textures, and are reasonably diverse. Textures have long been understood to have a particular value in training neural networks, and have recently been shown to be more informative than shapes (Geirhos et al., 2018). After about $10\%$ of the training run, most images produced by $G(\boldsymbol{z}; \phi)$ look like high frequency patterns that have little meaning to humans.

During training, the average probability of the class predicted by the teacher is about $0.8$. On the other hand, the most likely class according to student has an average probability of around $0.3$. These confidence levels suggest that the generator focuses on pseudo data that is close to the boundary decisions of the student, which is what we observed in the toy experiment of Figure 1. Finally, we also observe that the classes of pseudo samples are close to uniformly distributed during training, for both the teacher and student. Again this is not surprising: the generator seeks to make the teacher less predictable on the pseudo data in order to fool the student, and spreading its mass across all the classes available is the optimal solution.

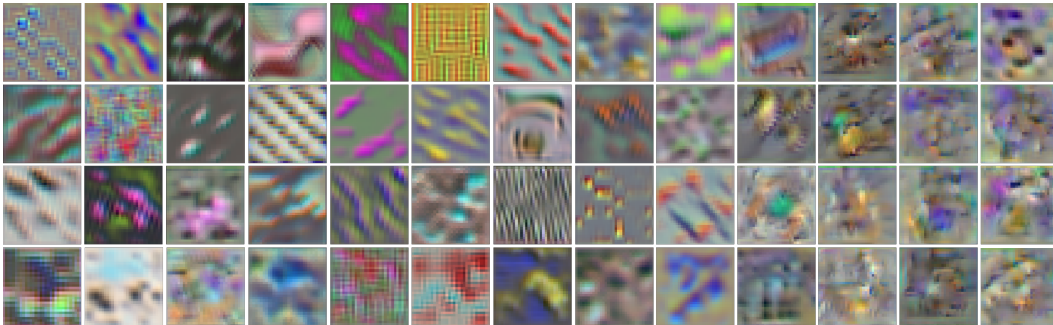

Figure 3: Pseudo images sampled from the generator across several seeds and hyperparameters. As the training progresses (left to right), pseudo data goes from coarse and diverse textures to complex high frequency patterns. Note that time is not to scale and most images look like the last four columns.

## 4.4 Measuring belief match near decision boundaries

Our understanding of the adversarial dynamics at play suggests that the student is implicitly trained to match the teacher's predictions close to decision boundaries. To gain deeper insight into our method, we would like to verify that this is indeed the case, in particular for decision boundaries near real images. Let $\mathcal{L}_{CE}$ be the cross entropy loss. In Algorithm 2, we propose a way to probe the difference between the beliefs of network $A$ and $B$ near the decision boundaries of $A$. First, we sample a real image $\boldsymbol{x}$ from the test set $\boldsymbol{X}_{test}$ such that network $A$ and $B$ both give the same class prediction $i$. Then, for each class $j \neq i$ we update $\boldsymbol{x}$ by taking $K$ adversarial steps on network $A$, with learning rate $\xi$, to go from class $i$ to class $j$. The probability $p_i^A$ of $\boldsymbol{x}$ belonging to class $i$ according to network $A$ quickly reduces, with a concurrent increase in $p_j^A$. During this process, we also record $p_j^B$, the probability that $\boldsymbol{x}$ belongs to class $j$ according to network $B$, and can compare $p_j^A$ and $p_j^B$. In essence, we are asking the following question: *as we perturb $\boldsymbol{x}$ to move from class $i$ to $j$ according to network A, to what degree do we also move from class $i$ to $j$ according to network B?*

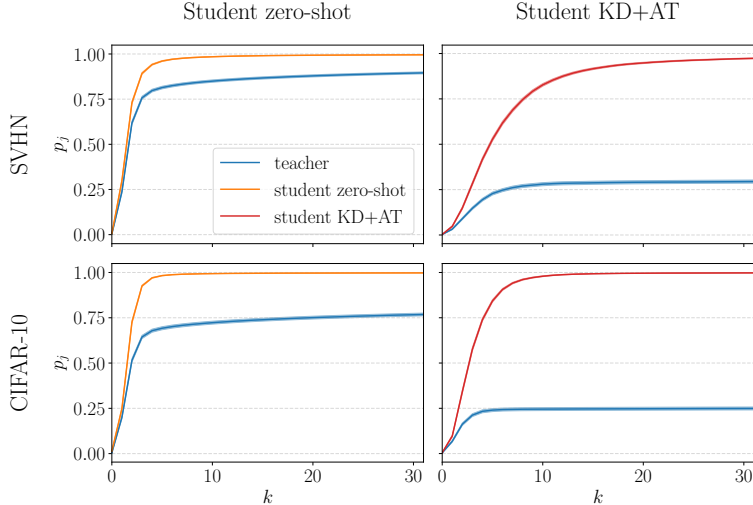

Figure 4: Average transition curves over 9 classes and 1000 images for both SVHN and CIFAR-10. LEFT, taking adversarial steps w.r.t. the zero shot student model, and RIGHT, w.r.t. the normal distilled student model. We note that the average of $p_j$ is much more similar between teacher and student when targeting the zero-shot boundary than when targeting the boundary of the student learnt via normal KD+AT distillation. Line thickness shows $\pm 2$ times the standard error of the mean.

We refer to $p_j$ curves as transition curves. For a dataset of $C$ classes, we obtain $C-1$ transition curves for each image $\boldsymbol{x} \in \boldsymbol{X}_{test}$, and for each network $A$ and $B$. We show the average transition curves in Figure 4, in the case where network B is the teacher, and network A is either our zero-shot student or a standard student distilled with KD+AT. We observe that, on average, updating images to move from class $i$ to class $j$ on our zero-shot student also corresponds to moving from class $i$ to class $j$ according to the teacher. This is true to a much lesser extent for a student distilled from the teacher with KD+AT, which we observed to have flat $p_j = 0$ curves for several images. This is particularly surprising because the KD+AT student was trained on real data, and the transition curves are also calculated for real data.

We can more explicitly quantify the belief match between networks $A$ and $B$ as we take steps to cross the decision boundaries of network A. We define the Mean Transition Error (MTE) as the absolute probability difference between $p_j^A$ and $p_j^B$, averaged over $K$ steps, $N_{test}$ test images and $C-1$ classes:

$$\text{MTE}(net_A, net_B) = \frac{1}{N_{test}} \sum_n^{N_{test}} \frac{1}{C-1} \sum_c^{C-1} \frac{1}{K} \sum_k^K \left| p_j^A - p_j^B \right| \tag{2}$$

The mean transition errors are reported in Table 2. Our zero-shot student has much lower transition errors, with an average of only 0.09 probability disparity with the teacher on SVHN as steps are taken from class $i$ to $j$ on the student. This is inline with the observations made in Figure 4. Note that we used the values $K = 100$ and $\xi = 1$ since they gave enough time for most transition curves to converge in practice. Other values of $K$ and $\xi$ give the same trend but different MTE magnitudes, and must be reported clearly when using this metric.

Table 2: Mean Transition errors (MTE) for SVHN and CIFAR-10, between our zero-shot student and the teacher, and between a student distilled with KD+AT and the teacher. Our student matches the transition curves of the teacher to a much greater extent on both datasets.

|          | Zero-shot (Ours) | KD+AT |
|----------|:----------------:|:-----:|
| SVHN     | 0.09             | 0.64  |
| CIFAR-10 | 0.22             | 0.68  |

# 5 Conclusion

In this work we demonstrate that zero-shot knowledge transfer can be achieved in a simple adversarial fashion, by training a generator to produce images where the student does not match the teacher yet, and training that student to match the teacher at the same time. On simple datasets like SVHN, even when the training set is large, we obtain students whose performance is close to distillation with the full training set. On more diverse datasets like CIFAR-10, we obtain compelling zero and few-shot distillation results, which significantly improve on the previous state-of-the art. We hope that this work will pave the way for more data-free knowledge transfer techniques, as private datasets likely become increasingly common in the future.

**Acknowledgements.** The authors would like to thank Benjamin Rhodes, Miguel Jaques, Luke Darlow, Antreas Antoniou, Elliot Crowley, Joseph Mellor and Etienne Toussaint for their useful feedback throughout this project. Our work was supported in part by the EPSRC Centre for Doctoral Training in Data Science, funded by the UK Engineering and Physical Sciences Research Council (grant EP/L016427/1) and the University of Edinburgh as well as a Huawei DDMPLab Innovation Research Grant. The opinions expressed and arguments employed herein do not necessarily reflect the official views of these funding bodies.

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
