[Supplementary Material]

# Supplementary Material for Zero-shot Knowledge Transfer via Adversarial Belief Matching

**Paul Micaelli**
University of Edinburgh
{paul.micaelli}@ed.ac.uk

**Amos Storkey**
University of Edinburgh
{a.storkey}@ed.ac.uk

## 1

Here we describe a number of loss terms that have been tried with our method in order to encourage some behaviour from the generator, but have all resulted in a decrease of performance from the student. This happens despite finding the optimal scaling for each loss term, denoted here by $\gamma$. In general, we believe that this is due to the generator already achieving the desired behaviours due to the nature of the adversarial dynamics, and so extra losses simply create an imbalance between the two adversaries. Extra loss terms added to $\mathcal{L}_G$ relate to:

1. **The entropy of the teacher**: $\mathcal{L}_G \mathrel{+}= \gamma \times \left( - \sum_i \boldsymbol{t}_p^{(i)} \log \boldsymbol{t}_p^{(i)} \right)$ where $\boldsymbol{t}$ are the teacher's probability outputs on pseudo data $\boldsymbol{x}_p$, and $i$ is the class index. When positive, this encourages the generator to search regions of the input space where the teacher is confident, which could correlate with regions close to the real data manifold.

2. **The entropy of the student**: $\mathcal{L}_G \mathrel{+}= \gamma \times \left( - \sum_i \boldsymbol{s}_p^{(i)} \log \boldsymbol{s}_p^{(i)} \right)$. This encourages the generator to take more risks and look for images that the student is confidently wrong about.

3. **The consistency of the images generated**: $\mathcal{L}_G \mathrel{+}= \gamma \times D_{KL}(T(\boldsymbol{x}_p) \,\|\, T(A(\boldsymbol{x}_p)))$ where $A$ is some augmentation operation, such as Gaussian noise or Gaussian blurring. Here the idea is to constrain the generator to search images for which being augmented does not change the output of the teacher. Again this is an attempt to drive the search closer to real data and away from adversarial images.

4. **The diversity of the images generated**: $\mathcal{L}_G \mathrel{+}= -\gamma \times \phi\phi^T$, where $\phi$ corresponds to the representation of a batch of images in the penultimate layer of the teacher. Here the loss encourages each batch to be diverse in the space spanned by the teacher's last layer. So at any one time, the generator is penalized if all of its samples look too similar according to the teacher.

On CIFAR-10 we observe that the volume each class occupies in the input space of common neural networks does not seem to be equal. Here, we produce uniform noise by sampling each pixel $x_i \sim U(0, 255)$ discretely, and normalizing the resulting images by the mean and standard deviations of CIFAR-10, as used during training time. The distribution of the predictions made by common neural networks (pretrained on CIFAR-10) are shown in Figure 1. The predictions are mostly birds or frogs, which suggests that decision boundaries have a higher density close to the real images. Another way to reason about this is that adding uniform noise to a real image is much more likely to change its class than adding uniform noise to uniform noise.

Figure 1: Distribution of predictions across different architectures when given 1000 images of uniform noise. Interestingly, the predictions are largely focused on two classes.