[Reviews · NeurIPS 2019]

Reviewer 1



[UPDATE] I am satisfied with the authors' reply about larger scale experiments, mentioned by 2 out of 3 reviewers. While I am only guessing that performance may degrade as a function of dataset scale, it is not hard to imagine advances in GANs which could make that degradation smaller, hence make the proposed method more useful. Further, even in an adversarial setting, it may be possible to guess what kind of inputs are relevant, or extend the method to few-shot or some hybrid approach. I am positively surprised that features of the student have comparable transferability to the teacher, I was concerned that some sort of overfitting to a teacher's decision boundary was possible, but this does not seem to be the case. While I agree with the authors that, in most cases, those releasing research models will not go out of their way to vaccinate them against zero-shot distillation, the proposed method could be used to (somewhat) copy and repurpose information stored in hardware model. Take for example Tesla's autopilot which uses several neural networks in it and is trained on tens of billions of images which are not available to the world. In fact, that data is the primary competitive advantage of Tesla in the long run and its acquisition costs Tesla millions of dollars. Since the autopilot needs to drive without an internet connection and inference needs to be extremely low latency, it is surely done on device and hackable. Your method could be used to train a good feature extractor comparable to that on their device. Further, it would not need to be zero-shot, since it's very easy to get a little bit of data. Overall I am impressed with the work and satisfied with the authors' response. Hence, I am increasing my score to 8. [OLD REVIEW] Originality: Highly original. Zero-shot distillation using some form of adversarial loss is imaginable, but the actual performance level is quite unexpected, imho. Quality: Good deltas to previous works, and error bars make the differences more convincing. Clarity: Writing is clear enough, although some more details are required for those who are only distantly familiar with the GAN literature. The supplementary materials are also on the light side. Significance: High. I believe this paper opens new avenues for investigation, basically offering a new attack vector against embedded inference devices.

Reviewer 2



The paper proposes an algorithm for distilling a teacher into a student without any data or metadata. The main idea is to train a generator to feed “hard examples” (where teacher and student disagree) to the student. The dynamics of this algorithm in Fig. 1 leads to an analysis that probes the difference between the predictions of the two networks near the decision boundaries in $4.4. Experiments are done on SVHN and CIFAR-10. 1. The paper is clear, concise, and generally well-written. One thing that could be improved is $4.1. The title does not reflect all the content. Also, can we highlight what the baselines are? Especially KD(+AT) (L172) as this is used many times. It would also be nice to remind the reader from time to time when discussing the results. 2. The paper is well-motivated (L27-39). 3. The approach is sound and intuitive. Possible conceptual concerns are explicitly mentioned ($3.4). I also appreciate that the authors mention the loss terms that have been tried but did not work (L112-117). 4. The paper gives the right context on the relevant work, helping the reader to quantify the significance of the proposed approach and experimental results. For instance, it makes sure to point out the concurrent work of Nayak et al. 2019 in L81-84 and then highlight that again in L218-221. 5. Solid experiments and analysis. Main results are shown in Fig. 2 and Table 1. Overall, I like that the experiments do not only attempt to showcase the effectiveness of the proposed approach but also to provide further insights including toy experiments ($3.3 and Fig. 1), the transition error in $4.4, and qualitative results ($4.3 and Fig. 3). ### Updated Reviews ### While the authors' statement regarding large-scale experiments seems fair, the rebuttal does not attempt to change my opinion regarding the paper's level of significance. I am happy to remain at a 7. Obviously, I would still recommend acceptance.

Reviewer 3



[Update after author feedback] I did not have any significant concerns with the paper originally, and the author feedback addresses the other reviewer's concerns about scaling to large datasets. I continue to recommend acceptance. This paper provides a method called zero-shot KT for performing distillation from a teacher model to a smaller student model in the zero-shot setting (i.e. the student has no access to the teacher's training data). An adversarial generator is trained alongside the student and seeks to generate pseudo-examples that maximize the KL-divergence in the class predictive distributions between the teacher and student. The student seeks to minimize this same KL-divergence. The proposed model is shown to be competitive with or outperform distillation methods that have access to significant amounts of training data. In addition, the paper proposes a metric to quantify the mismatch between teacher and student as images from the test dataset are adversarially attacked. As far as I can tell, this work is original. Rather than relying on auxiliary information about classes, the proposed zero-shot KT approach only relies on the teacher model itself. Related work is detailed enough to clearly understand the novelty of the work and how previous approaches are related. The quality of this work is high. Figure 1 demonstrating results on a toy problem and figure 3 showing adversarially generated pseudo-examples were helpful inclusions for understanding the method. In Table 1, the authors show the circumstances under which zero-shot KT is outperformed by KD+AT, albeit KD+AT having more examples. Results were compared over a range of different architectures and over 3 initial seeds. The authors also addressed potential concerns head-on, such as issues with the adversary and choosing hyperparameters. Clarity is excellent. The writing was clear and insightful. The method and hyperparameters were discussed in sufficient detail to reproduce the results. Significance is high as well. As the authors point out in the introduction, this method is useful for distilling and making accessible pre-trained models for which the dataset may not be released due to intellectual property, privacy, size, or other concerns. Such scenarios will only become more likely in the future. A potential concern is the choice of hyperparameters, but the authors describe how they chose the hyperparameters by using another task and that the hyperparameters do not have a large effect on performance. The teacher transition curves in Figure 4 was a point of confusion for me. I was under the impression that the teacher network was the one being attacked and so was surprised that the performance of the teacher differed so drastically between the left and right columns. Is there any particular reason why the student model was the one attacked and not the teacher?

[Author Response · NeurIPS 2019]

We would like to thank all the reviewers for their time and for giving our paper this encouraging feedback.

**Reviewer #1** mentioned the issue of transferability of the student features when trained without any real data, which is also something we consider interesting. Since submission we tested our student on the ImageNet component from the CINIC-10 dataset (https://arxiv.org/abs/1810.03505), corresponding to a subset of ImageNet images chosen and resized to match CIFAR. We observe that the student generalizes equally well to its teacher, i.e. that the difference in performance between teacher and student remains the same under such dataset shift. This is not as much of a shift as transferring to a segmentation task (in general classification features generalize poorly to segmentation tasks even in normal circumstances). But in general we find that the student's representation shows similar transferability to that of the teacher, because of the student's tendency to directly match the decision boundaries of the teacher. However we cannot make any claim of increased transferability at this stage.

Vaccination is also an interesting thought. In practice the nature of the problem forces the student to match the teacher on the entire input space regardless of how it was trained, but having a teacher agnostic to out of distribution data could reduce the task-relevant information that leaks out to the student. On the other hand, depending on the vaccination approach it could be that the agnosticism of the teacher helps identifying samples close to the training data more easily, because they would have a specific entropy signature (high confidence). We have not pushed further on vaccination at this stage because it seems to be a use-case that is harder to justify in practice: someone releasing a trained model for people to use typically wouldn't take measures against people compressing it.

We agree with the reviewers that additional large scale datasets would be complementary to this work. However we also note that many of the competing papers we cited are limited to MNIST experiments due to the difficulty of the problem, so our tests are more extensive than many. In practice, a limitation to our current model is computational cost, since our distillation trains two networks instead of one, and several gradient steps per batch are taken for the student. This amounts to a handful times more computational cost than training a vanilla network, and our limited computational resources have made ImageNet scale experiments a challenge. This is a focus of future work; however those working in this field need to know of methods that are effective on the smaller problems so we can focus our efforts on scale in the right direction. Hence we think it is important to not wait for huge-scale experiments before releasing this work.

In particular in future work we plan to investigate more closely the relationship between the size of the teacher's training set and the difficulty of zero-shot distillation. Extra challenges relating to large scale datasets may include the larger number of classes, some of which may not be sampled enough by the generator.

We thank **Reviewer #2** for his comments on layout, and have modified our paper in light of that for the camera ready version.

**Reviewer #4** correctly and pertinently noticed that in Figure 4 we took adversarial steps on the students rather than the teacher. Attacking the students rather than the teacher allows us to show 4 curves per dataset instead of 3, and as such it conveys a bit more information. For instance it shows that it is easier to cross decision boundaries on our zero-shot student than on the baseline student when stepping on each network respectively (orange vs red curve per row). Another possible motivation comes from considering the case that the teacher and student are sets of linear decision boundaries. If we assume that the students is a smaller set (less capacity) mostly contained within the teacher set, then crossing decision boundaries of the student should correspond to crossing decision boundaries of the teacher, to a greater extent than the other way around.

Lastly we note that the batchnorm layers of the WRNs in our paper were initially not trained (gamma and beta fixed). This was a mistake because a trainable batchnorm is the most common practice for WRNs in the literature, and it facilitates the training procedure. We re-ran all the experiments in the paper and obtained a 1 to 2% boost to most of our accuracies, and reduced the Transition Error (TE) on CIFAR-10 by a third. For the sake of reproducibility we will ensure the camera ready version has these updated values, so that all values match our latest published code.

Again we thank the reviewers for their time and hope that we have answered their questions.

[Meta-Review · NeurIPS 2019]

All reviewers appreciated the work and recommend acceptance. The authors are encouraged to address the reviewers comments and include the author response in the camera-ready version of the manuscript.